# Tin prefiltration in computed tomography does not significantly alter radiation-induced gene expression and DNA double-strand break formation

Simone Schüle[1,2,3], Carsten Hackenbroch[2,3], Meinrad Beer[3], Patrick Ostheim[1,4], Cornelius Hermann[1], Razan Muhtadi[1], Samantha Stewart[1], Matthias Port[1], Harry Scherthan[1], Michael Abend[1]*

1 Bundeswehr Institute of Radiobiology Affiliated to the University of Ulm, Munich, Bavaria, Germany, 2 Department of Diagnostic and Interventional Radiology and Neuroradiology, German Armed Forces Hospital of Ulm, Ulm, Baden-Württemberg, Germany, 3 Department of Radiology, University Hospital of Ulm, Ulm, Baden-Württemberg, Germany, 4 Department of Radiology, University Hospital Regensburg, Regensburg, Bavaria, Germany

* michaelabend@bundeswehr.org

**Data Availability Statement:** All relevant data are within the manuscript and its Supporting Information files.

## Abstract

### Background

The tin (Sn) prefilter technique is a recently introduced dose-saving technique in computed tomography (CT). This study investigates whether there is an altered molecular biological response in blood cells using the tin prefiltering technique.

### Methods

Blood from 6 donors was X-irradiated *ex-vivo* with 20 mGy full dose (FD) protocols (Sn 150 kV, 150 kV, and 120 kV) and a tin prefiltered 16.5 mGy low dose (LD) protocol on a CT scanner. Biological changes were determined by quantification of γH2AX DNA double-strand break (DSB) foci, and differential gene expression (DGE) relative to unexposed samples were examined for seven known radiation-induced genes (*FDXR*, *DDB2*, *BAX*, *CDKN1A*, *AEN*, *EDA2R*, *APOBEC3H*) and 667 microRNAs (miRNA).

### Results

*EDA2R* and *DDB2* gene expression (GE) increased 1.7-6-fold (p = 0.0004–0.02) and average DNA DSB foci value (0.31±0.02, p<0.0001) increased significantly relative to unexposed samples, but similarly for the applied radiation protocols. *FDXR* upregulation (2.2-fold) was significant for FD protocols (p = 0.01–0.02) relative to unexposed samples. miRNA GE changes were not significant (p = 0.15–1.00) and DGE were similar for the examined protocols (p = 0.10–1.00). An increased frequency of lower DGE values was seen in the Sn 150 kV LD protocol compared to the 120 kV FD and Sn 150 kV FD protocols (p = 0.001–0.008).

**Funding:** The author(s) received no specific funding for this work.

**Competing interests:** The authors have declared that no competing interests exist.

## Conclusions

The current *ex-vivo* study indicates no changes regarding transcriptional and post-transcriptional DGE and DNA DSB induction when using the tin prefilter technique and even a significant tendency to lower radiation-induced DGE-changes due to the dose reduction of the tin prefilter with equal image quality compared to classical CT scan protocols was found.

## Introduction

The increasing number of annual computed tomography (CT) examinations demand dose reduction methods to keep medical radiation absorbed doses to the population as low as reasonably achievable (ALARA) [1, 2]. Over the past two decades, numerous dose-saving techniques have been introduced in CT scanners, ranging from improved hardware, such as more efficient detector technology or tin (Sn) prefiltration, to software applications, such as iterative reconstruction algorithms [3–5]. Tin prefiltration is one of the last innovations introduced in the third-generation dual-energy CT scanners with energy-integrating detectors. It is also integrated into the CT scanner with the first-generation photon-counting detector. The < 1 mm thick tin prefilter is placed directly downstream of the X-ray tube, absorbing the low-energy X-rays and consecutively resulting in a higher energetic X-ray spectrum before it crosses the patient's body. The mean photon energy of a 120 kV and 150 kV protocol is 64.2 keV and 72.1 keV, respectively, while the mean photon energy of a 150 kV protocol with tin prefiltration is 96.6 keV [6]. The shifted X-ray spectrum consists of more high-energy photons that are better able to penetrate and thus depict dense areas of the body, such as the shoulder girdle or body regions with metal implants [7–9]. The high-energy X-ray spectrum allows the applied radiation dose to be reduced without compromising image quality [10, 11]. In addition, with low-energy photons, most of their energy is deposited locally, thus contributing to the patient's radiation dose. By filtering out the low-energy photons by the tin prefilter, the patient's radiation dose is further reduced. However, as a result, soft tissue contrast is reduced by using the tin prefilter technique. For high contrast examinations, such as paranasal [10, 12–14] and pulmonary imaging [15, 16] soft tissue contrast is not as important. Also for kidney stone detection CTs [17, 18] and calcium scoring in cardiac CTs [19], tin prefilter protocols are meanwhile well established.

In the present study, we investigated the molecular biological effects of tin prefiltration using two well-known assays that can measure radiation-induced changes at the molecular level, namely the γH2AX DNA double-strand break (DSB) and the gene expression (GE) assay with which, in our hands, radiation doses as low as 4 mGy and 5 mGy, respectively, can be detected [20, 21]. The following four questions were addressed: Are radiation-induced molecular biological changes 1) detectable, 2) altered due to tin prefiltration, 3) lowered due to tin prefiltration-induced dose reduction? And, can current radiobiological assays detect a 5 mGy dose reduction?

## Material and methods

### Sample collection and irradiation

For transcriptome (mRNA) and γH2AX DNA DSB analysis, peripheral whole blood from 6 healthy donors (two female and four male volunteers, mean age: 36 ± 8 years) was collected into S-Monovette® 4.9 ml EDTA tubes (Sarstedt AG & Co. KG, Nümbrecht, Germany). For

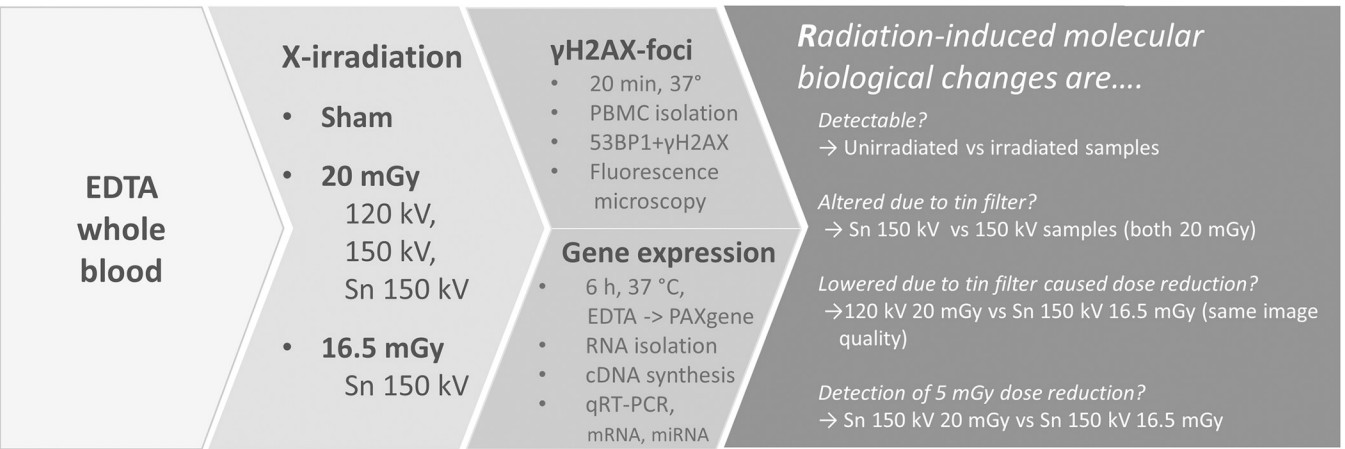

**Fig 1. Study overview.** Study overview including X-irradiation of EDTA whole blood in a 3rd generation dual-source computed tomography scanner, incubation time of samples, and workflow of gene expression and γH2AX DNA DSB foci measurements and the addressed research questions of this study. The 53BP1 was utilized for methodological purposes. Abbreviations, cDNA = complementary deoxyribonucleic acid, CT = computed tomography, DSB = double-strand break, EDTA = ethylenediaminetetraacetic acid, GE = gene expression, kV = kilovolt, n = number, PBMC = peripheral blood mononuclear cells, qRT-PCR = quantitative real-time polymerase chain reaction, RNA = ribonucleic acid.

post-transcriptome (miRNA) analysis, peripheral whole blood from 5 healthy donors (two female and three male volunteers, mean age: 36 ± 9 years) was used. The following inclusion criteria apply to the healthy donors: age ≥ 18 years, no acute health problems, no previously known diseases of the haematopoietic system, signed written informed consent form, ability to give consent, naivety towards exposure to higher radiation doses (< 1 mSv/life year). Blood tubes were either sham irradiated, irradiated at 20–22 mGy with a 150 kV, a Sn 150 kV, and a 120 kV spectrum, or irradiated at 16.5 mGy with a Sn 150 kV spectrum in terms of dose in water (Fig 1). The radiation dose of the Sn 150 kV 16.5 mGy protocol was chosen so that the image quality of the Sn 150 kV 16.5 mGy CT images was equivalent to that of the 120 kV 20 mGy CT images. For this purpose, the image quality was assessed objectively. The standard deviation of the mean Houndsfield Unit value of a region of interest (ROI) in the air in the CT image was used as the image quality parameter (background noise). The radiation dose of the Sn 150 kV LD protocol was determined as follows: First, the image noise of the 120 kV FD protocol was measured with triplicate ROI measurements in CT images, then the radiation dose (mAs) of the Sn 150 kV protocol was lowered until triplicate ROI measurements in CT images at the same position showed the same image quality as the 120 kV FD protocol. *Ex-vivo* irradiation (Table 1) was performed at 37˚C in a water phantom (S1 Fig.) with a 32 cm diameter

**Table 1. Overview of CT scan acquisition parameters per examined scan protocol.**

| Scan Protocol | Dose* (mGy) | CTDIvol (mGy) | DLP (mGy*cm) | Dose rate (mGy/s) | Tube current (mAs) | Mean photon energy (keV) | Std.Dev. ROI Air (HU) |
|---|---|---|---|---|---|---|---|
| 150 kV FD | 20.0 ± 1,0 | 22.9 | 359 ± 0 | 65.6 ± 0.09 | 203 | 72.1 | 22.3 ± 1.4 |
| Sn 150 kV FD | 21.8 ± 0.8 | 21.2 | 332 ± 0 | 60.6 ± 0.13 | 728 | 98.6 | 20.3 ± 1.5 |
| Sn 150 kV LD | 16.5 ± 1.0 | 16.5 | 261 ± 0 | 47.5 ± 0.06 | 570 | 98.6 | 21.8 ± 0.4 |
| 120 kV FD | 19.9 ± 1.1 | 24.2 | 379 ± 0 | 69.1 ± 0.15 | 361 | 64.2 | 22.5 ± 1.5 |

All protocols were scanned with the following parameters: collimation: 192 x 0.6 mm, pitch: 0.6, rotation time: 1/s, and a scan time of 5.5 s. The asterisk marks the mean radiation dose measured with thermoluminescence dosimeters ± standard deviation (Kerma in water). Abbreviations, CTDIvol = computed tomography dose index, DLP = dose length product, FD = full dose, kV = kilovolt, LD = low dose, ROI = region of interest, Sn = tin prefilter, Std.Dev. = standard deviation.

using a 3rd-generation dual-source CT scanner (Somatom Force, Siemens Healthineers, Erlangen, Germany). In short, all samples were scanned with protocols with a scan time of 5.5 s, a collimation of 192 x 0.6 mm, a rotation time of 1/s, and a pitch of 0.6. To examine background noise, image reconstructions were performed with Admire level 2, a slice thickness of 2 mm, an increment of 2, a Br59 kernel, and *in osteo* window. The applied radiation dose was measured with thermoluminescence dosimeters (TLD) (Karlsruhe Institute of Technology, Karlsruhe, Germany) in terms of kerma in water. The TLDs were placed next to the blood samples in the water phantom for each CT scan.

For the GE analyses after irradiation, whole blood was incubated in EDTA tubes for 6 h at 37˚C to allow for radiation-induced gene expression changes. The 6 h incubation time was chosen to simulate an *in-vivo* irradiation as a compromise based on previous *ex-vivo* experiments with human peripheral whole blood culturing [22]: *Ex-vivo* incubation over 4–6 h represents a sufficient biological response time, and GE changes comparable to the *in-vivo* situation were observed for the examined genes. However, at 12 h and later, a massive decrease in cell numbers and reduction in isolated RNA amount, even in unexposed blood samples, indicates insufficient culture conditions and that a limited *ex-vivo* culture time needs to be considered in such experiments [23]. After this incubation time, 2.5 ml of whole blood was transferred into a PAXgene® Blood RNA tube (BD Diagnostics, PreAnalytiX GmbH, Hombrechtikon, Switzerland) to avoid RNA degradation and to "freeze" the gene expression response. The samples were kept at room temperature for at least 2 h and stored at -20˚C until further processing.

For the γH2AX DNA DSB focus analyses, whole blood samples were incubated in EDTA tubes for 20 min at 37˚C after irradiation. This time point after irradiation represents a well-accepted standard for performing *ex-vivo* experiments using this endpoint [21, 24, 25]. Peripheral blood mononuclear cells (PBMC) were separated from blood samples by the Ficoll Paque Plus (Merck, Darmstadt, Germany) density centrifugation (1000g 10 min), washed twice with the Roswell Park Memorial Institute (RPMI) 1640 Medium (GIBCO™, Life Technologies, Darmstadt, Germany), fixed with 70% ethanol and stored at -20˚C until further processing.

Ethical approval was obtained from the local ethics committee of the University of Ulm, Germany (No.301/20). Written informed consent was obtained from each healthy blood donor prior to blood collection. The recruitment period in which the samples for this study were collected began on 01.08.2021 and ended on 22.08.2021.

## Immunostaining and DNA DSB focus analysis

Immunofluorescence staining and DNA DSB focus analysis were performed as described previously [21, 26, 27]. To this end co-localizing γ-H2AX + 53BP1 foci were considered as surrogate markers for DNA DSBs and counted in 100 well-separated and morphologically intact PBMC nuclei by an experienced investigator (HS) using a Zeiss Axioimager Z2 epifluorescence microscope (Zeiss, Oberkochen, Germany) of an ISIS fluorescence imaging system (MetaSystems, Altlussheim, Germany) equipped with appropriate single and dual-band filter sets. Radiation-induced foci (RIF) were calculated by subtracting the counted foci per cell of the respective unexposed sample from the counted foci per cell value of the irradiated sample of each donor [20]. A RIF of zero indicates return to a value of the respective unexposed samples.

## Gene expression analyses

**RNA extraction and quality control.** RNA from PAXgene® Blood RNA tubes was isolated manually following a combination of the PAXgene Blood RNA system protocol (BD

Diagnostics, PreAnalytiX GmbH, Hombrechtikon, Switzerland) and the mirVana™ kit protocol (Invitrogen™, ThermoFisher Scientific, Carlsbad, CA 92008; USA/Life Technologies, Darmstadt, Germany). First, the PAXgene® Blood RNA samples were thawed, washed, and centrifuged according to the PAXgene Blood RNA system protocol (BD Diagnostics, PreAnalytiX GmbH, Hombrechtikon, Switzerland). After the protein digestion (Proteinase K; BD Diagnostics, PreAnalytiX GmbH, Hombrechtikon, Switzerland), the Lysis/Binding Solution was added, and further steps were performed according to the mirVana kit protocol (Invitrogen™, ThermoFisher Scientific, Carlsbad, CA 92008; USA/Life Technologies, Darmstadt, Germany). In brief, the total RNA, including small RNA species, was isolated by combining a Phenol–Chloroform RNA precipitation with further processing and purification using a silica membrane. After several washing procedures to purify RNA from other residual debris, DNA remnants became digested on the membrane (RNAse free DNAse Set, Qiagen, Hilden, Germany). We eluted the RNA with 100 μl of preheated RNAse free water (95°C) and stored the samples at -20°C until further examined quantitatively and qualitatively.

The isolated RNA samples were quantified spectrophotometrically (NanoDrop™, PeqLab Biotechnology, Erlangen, Germany). RNA integrity was assessed by the 4200 TapeStation System or the 2100 Agilent Bioanalyzer (both Agilent Technologies, Santa Clara, USA). Possible contamination due to genomic DNA was checked by running a PCR using primers for the actin gene, followed by gel electrophoresis. RNA specimens with a ratio of A260/A280 nm $\geq$ 2.0 and RNA integrity numbers (RIN) $\geq$ 7 were processed for qRT-PCR analysis.

**Transcriptional examinations using Real-Time Quantitative Reverse Transcription Polymerase Chain Reaction (qRT-PCR).** Aliquots of total RNA (0.5 μg) were reverse transcribed via the High-Capacity cDNA Reverse Transcription Kit (Applied Biosystems™, Life Technologies, Darmstadt, Germany). The qRT-PCR reactions were performed using commercially available and custom made TaqMan assays (*FDXR (AR7DTG3)*, *DDB2 (AR47X6H)*, *AEN (Hs00224322_m1)*, *BAX (Hs00180269_m1)*, *CDKN1A (Hs00355782_m1)*, *APOBEC3H (APYMRHF)*, and *EDA2R (Hs00939736_m1)*). PCR reactions were performed for *FDXR* and *DDB2* with additional TaqMan assays, which are either our in-house standard or had been shown to be particularly suitable as biomarkers for radiation exposure in a previous study of ours following high-dose radiation exposure (*FDXR (Hs01031621_g1, Hs00244586_m1, Hs01031617_m1)*, *DDB2 (Hs03044951_m1, Hs00172068_m1)* from [28]). Herefore, equal amounts of the template cDNA (*FDXR*, *DDB2*, *AEN*, *BAX*, *CDKN1A*, and *APOBEC3H* and 50 ng per reaction for *EDA2R*) were mixed with the TaqMan® Universal PCR Master Mix and run using a QuantStudio™ 12K OA Real-Time PCR System (Thermo Fisher Scientific Inc., Waltham, USA). The cycle threshold (Ct) values of the genes were normalized relative to diluted (0.01 ng per reaction) 18S rRNA (Hs99999901_s1). A manual threshold of 0.05 was set. For each donor the ratio (the differential gene expression, DGE) relative to the unexposed sample was determined by the ΔΔCt approach (DGE = $2^{-\Delta\Delta Ct}$) [29]. A DGE of one corresponds to a gene expression like unexposed samples. A DGE higher or lower than one refers to a several-fold over- or under-expression of the gene of interest after exposure relative to the reference. All experimental work was performed according to the standard operating procedures implemented at our laboratory since 2008 when the Bundeswehr Institute of Radiobiology became DIN-certified by TÜV Süd München, Germany (DIN EN ISO 9001/2008).

**Post-transcriptional analysis on 667 microRNAs (miRNA).** A commercially available 384-well low-density array (LDA) was used to measure miRNAs and to detect 380 different miRNAs on all RNA samples simultaneously. Two different LDAs (type A and B) were combined so that the detection of 667 miRNA species (partly spotted in duplicate to fill the LDA) from currently 1917 reported miRNA (miRbase v22) was possible. Aliquots from each RNA sample (in general, 2 μg total RNA/LDA type A/B) were reversely transcribed without

preamplification over 3 h using the "Megaplexe pools without preamplification" protocol for microRNA expression analysis. Using different sets of primers, two kinds of cDNAs suitable for each of both LDAs were created. The whole template cDNA and 450 µl 2x RT-PCR master mix were adjusted to a total volume of 900 µl by adding nuclease-free water, and aliquots of 100 µl were pipetted into each fill port of a 384-well human LDA. Cards were centrifuged twice (see above), sealed, and transferred into the 7900 qRT-PCR instrument. The 384-well LDA qRT-PCR protocol was run over 2 h. For normalization purposes, the median miRNA expression was used on each LDA since this method was more robust and slightly more precise compared to use of a housekeeping miRNA species provided on the LDA. The Ct values of the housekeeping gene were subtracted from the Ct value of each of the spotted genes, following the ΔCt quantitative approach for normalization purposes. The unexposed samples were chosen as references for DGE calculation employing the ΔΔ-Ct approach.

## Statistical analysis

Descriptive statistics were performed using Excel (Microsoft, Redmond, United States). Results were presented as mean value ± standard deviation or mean value ± standard error of the mean. Group comparisons were performed using either parametrical (Student's t-test, Welch's t-test) or non-parametrical tests (Mann-Whitney Rank Sum Test), where applicable. The frequency distribution of gene expression data was tested using the Chi-square test (SAS, SAS Institute Inc., Cary, United States). The graphical representations were performed using PowerPoint (Microsoft, Redmond, United States) and SPW (SigmaPlot, Version 14.5, Jandel Scientific, Erkrath, Germany).

## Results

### DNA double-strand break analysis

**Are radiation-induced biological changes detectable?.** Relative to unexposed samples, a significant increase ($p \leq 0.001$) in RIF values were observed for all examined protocols (Fig 2 and S1 Table). Mean RIF-values increased from 0.00 +/- 0.04 in unexposed samples to a mean value of 0.31 ± 0.02 (averaged over all CT exposure protocols, Fig 2 and S2 Fig).

**Are radiation-induced biological changes altered due to tin prefilter?.** The results revealed that application of the tin prefilter had no significant (150 kV FD vs. Sn 150 kV FD, $p = 0.53$) impact on formation of radiation-induced DNA DSBs.

**Are radiation-induced biological changes lowered due to tin prefilter, and is a 5 mGy dose reduction detectable?.** A 20% dose reduction (~3.5 mGy) at the same image quality (120 kV FD vs. Sn 150 kV LD) and a 5 mGy dose reduction (Sn 150 kV FD vs. Sn 150 kV LD) had no significant impact on the number of radiation-induced DNA DSBs (p = 0.31–0.98).

In addition, a frequency distribution analysis was performed for the three comparisons (150 kV FD vs. Sn 150 kV FD, 120 kV FD vs. Sn 150 kV LD, and Sn 150 kV FD vs. Sn 150 kV LD), classifying the results in higher and lower RIF categories. The frequency distribution within these categories was random for all comparisons (Chi-square $p$ = 0.08–0.56, S3 Fig).

### Transcriptional qRT-PCR measurements on seven known radiation-induced genes

**Are radiation-induced biological changes detectable?.** For *DDB2* and *EDA2R*, significant ($p = < 0.0001$–0.01) DGE changes were detectable in blood samples of all examined CT exposure protocols (Fig 3 and S1 Table) relative to unexposed samples. For *DDB2*, DGE ranged from 0.96 to 3.01 with a mean value of 1.72 ± 0.09, and for *EDA2R*, from 3.39 to 10.94 with

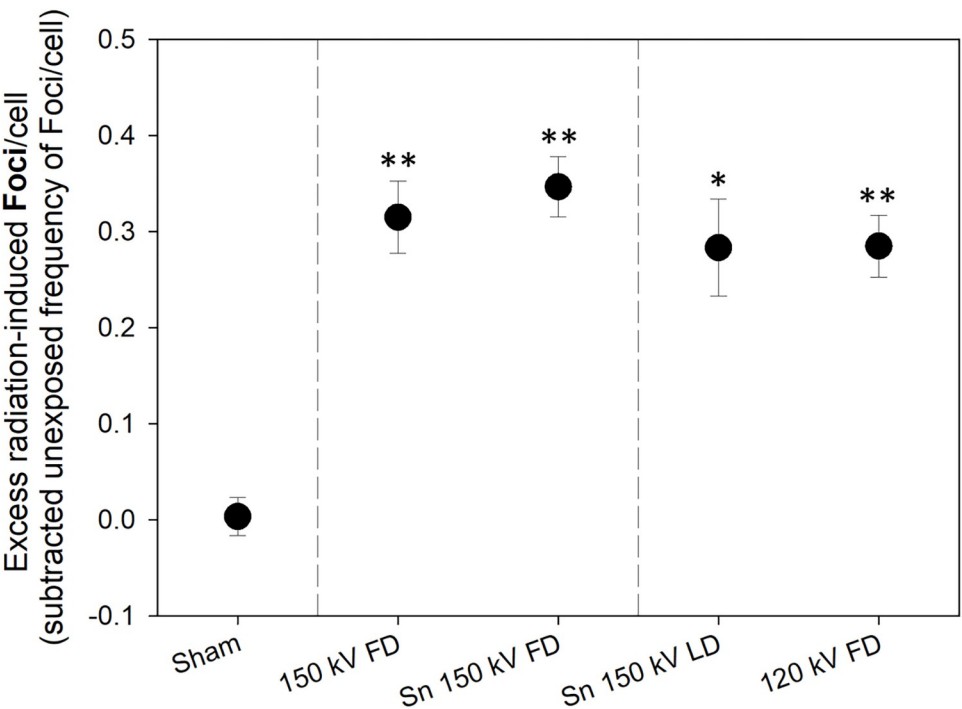

**Fig 2. Results of DNA double-strand break analysis.** Scatter plot of excess (subtracted spontaneous Foci/cell) radiation-induced Foci per cell for each examined CT scan protocol are shown, leading to, on average, 0 Foci in Sham (un-)exposed samples. The symbols reflect mean values, and the error bars represent the standard error of the mean, which is n = 6. P-values <0.001–0.0001 and <0.0001 are marked with one and two asterisks and refer to significant differences relative to unexposed values. Abbreviations: FD = full dose, kV = kilovolt, LD = low dose, Sn = tin prefilter.

a mean value of 5.90 ± 0.42. For *FDXR* significant (*p* = 0.01–0.02) GE changes were only detectable for FD protocols, and DGE ranged from 0.85 to 3.82 with a mean value of 2.21 ± 0.18. In the LD protocol, there was an insignificant (*p* = 0.05) *FDXR* increase, and DGE ranged from 0.72 to 2.86 with a mean value of 1.86 ± 0.26. In *AEN* and *BAX*, only the Sn 150 kV FD and 120 kV FD resp. the Sn 150 kV FD protocol showed significant (*p* = 0.02–0.04) GE changes with DGE ranging from 0.90 to 2.81 and a mean value of 1.86 ± 0.15 for *AEN* and with DGE ranging from 0.87 to 2.36 and a mean value of 1.73 ± 0.21 for *BAX*. *CDKN1A* and *APOBEC3H* showed insignificant (*p* = 0.08–0.94) radiation-induced GE changes with DGEs ranging for *CDKN1A* from 0.79 to 3.75 and a mean value of 1.55 ± 0.13 and for *APOBEC3H* ranging from 0.57 to 4.51 with a mean value of 1.67 ± 0.25.

**Are radiation-induced biological changes altered due to tin prefilter?.** The application of tin prefiltration did not significantly (*p* = 0.19–0.70) change radiation-induced GE levels in all gene studies when comparing the GE of 150 kV FD with Sn 150 kV FD protocols. In addition, a frequency distribution of lower resp. higher GE level per sample was performed only for those genes (*FDXR, DDB2, EDA2R*) in which radiation-induced changes were detectable in the majority of samples (Fig 4A). The frequency distribution showed no significant (Chi-square *p* = 0.05) difference between 150 kV FD and Sn 150 kV FD samples.

**Are radiation-induced biological changes lowered due to tin prefilter and is a 5 mGy dose reduction detectable?.** There is no significant (*p* = 0.14–0.69) GE difference in all genes studied of the 120 kV FD compared to the Sn 150 kV LD protocol with a 20% dose reduction. However, the frequency distribution between Sn 150 kV LD and 120 kV FD was not random (Chi-square p = 0.001), with the majority of DGE induced by Sn 150 kV LD samples falling into the low fold-change category (Fig 4B).

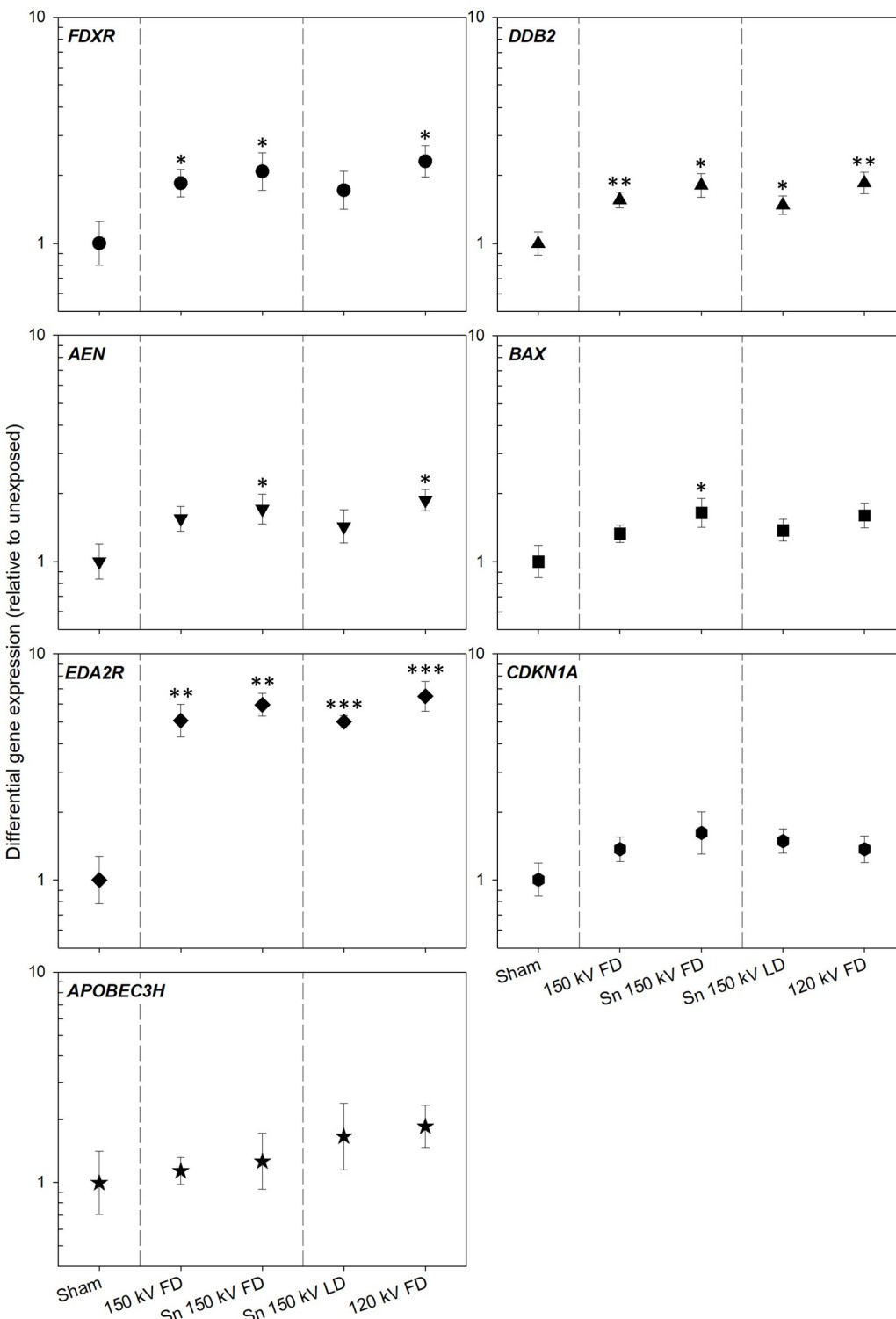

**Fig 3. Results of transcriptional qRT-PCR measurements.** Scatter plot of differential gene expression of *FDXR*, *DDB2*, *AEN*, *BAX*, *EDA2R*, *CDKN1A*, and *APOBEC3H* (relative to unexposed) for each examined CT scan protocol. The symbols reflect mean values, and error bars represent the standard error of the mean. P-values <0.05–0.01, <0.01–0.001, and <0.001–0.0001 are marked with one, two, and three asterisks and refer to significant differences relative to unexposed values. Abbreviations: FD = full dose, kV = kilovolt, LD = low dose, Sn = tin prefilter.

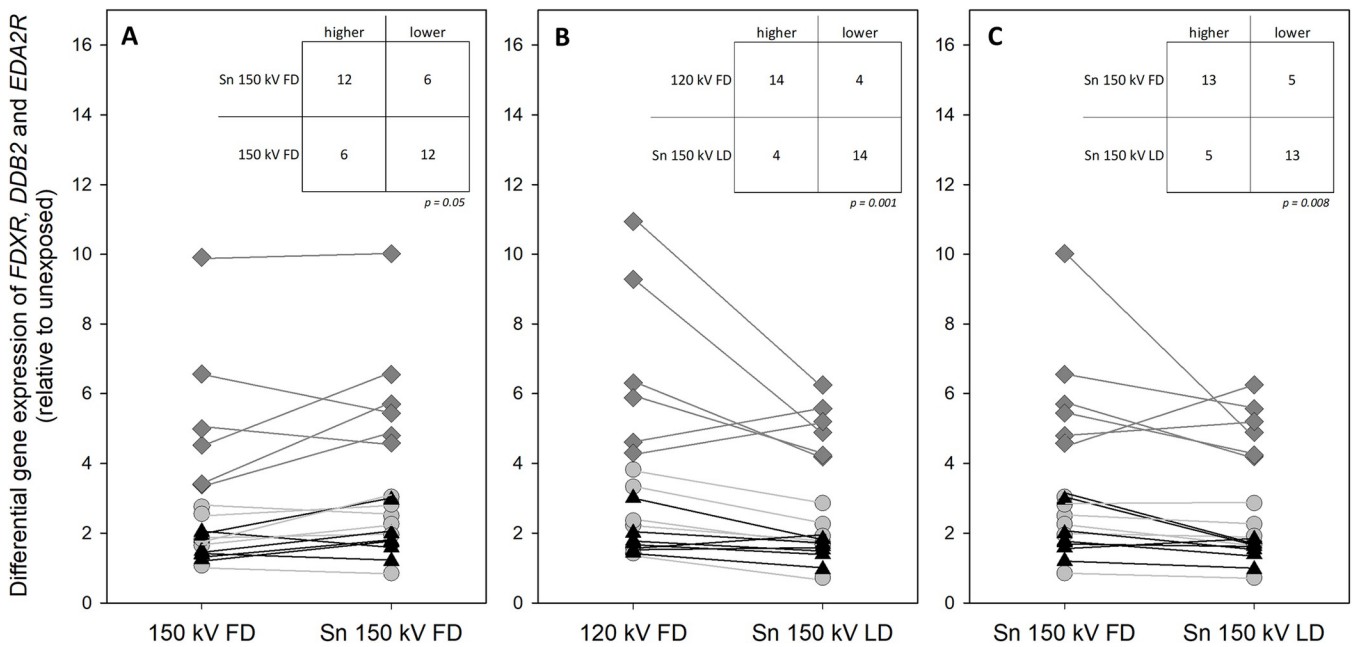

**Fig 4. Frequency distribution results of differentially expressed genes.** Graphical (connected scatter plot) and tabular (chi-square table) representation of the frequency distribution of higher resp. lower radiation-induced GE changes of *FDXR (grey circle)*, *DDB2*▲, and *EDA2R (grey diamond)* for the following research questions: A) 150 kV FD vs. Sn 150 kV FD, B) 120 kV FD vs. Sn 150 kV LD and C) Sn 150 kV FD vs. Sn 150 kV LD. The symbols reflect the mean values of DGE per category. The line connects the DGE results of the same donor for both compared groups. Abbreviations: FD = full dose, kV = kilovolt, LD = low dose, Sn = tin prefilter.

No significant change ($p = 0.21–0.56$) in GE levels was seen in all genes studied when comparing the Sn 150 kV FD with the 5 mGy lesser X-irradiated Sn 150 kv LD protocol. Again, the frequency distribution between Sn 150 kV LD and Sn 150 kV FD was not random (Chi-square $p = 0.008$), with the majority of DGE induced by Sn 150 kV LD samples falling into the low fold-change category (Fig 4C).

The results did not differ when different Taq-Man assays were used for *FDXR* and *DDB2* (S4 Fig).

**Post-transcriptional measurements on altogether 667 different miRNA species using LDAs.** In the first step, all plotted miRNAs on the LDAs were analysed according to Fig 5 (upper half), which resulted in 48 miRNAs that showed a radiation-induced DGE increase $\geq$ |1.5 | in the samples of at least one of the examined protocols. In the second step, the four addressed research questions (Fig 1) were analysed separately (Fig 5, lower half).

**Are radiation-induced biological changes detectable?.** Here, we found 8 miRNAs that showed a radiation-induced up- or down-regulation in all FD X-irradiated samples. However, all detected DGE changes were non-significant ($p = 0.15–1.00$, S2 Table). Since no significant radiation-induced dysregulation of miRNA was detected, a frequency distribution analysis was not performed.

**Are radiation-induced biological changes altered due to tin prefilter?.** Application of the tin prefilter did not result in any significant ($p = 0.01–1.00$) DGE change in the 22 miRNAs that showed up- or down-regulation in either the 150 kV FD or Sn 150 kV FD samples.

**Are radiation-induced biological changes lowered due to tin prefilter and is a 5 mGy dose reduction detectable?.** Also, there is no significant ($p = 0.12–1.00$) DGE difference between the 120 kV FD compared to the Sn 150 kV LD protocol with a 20% dose reduction in the 41 miRNAs, that showed up- or down-regulation in either the 120 kV FD or Sn 150 kV LD

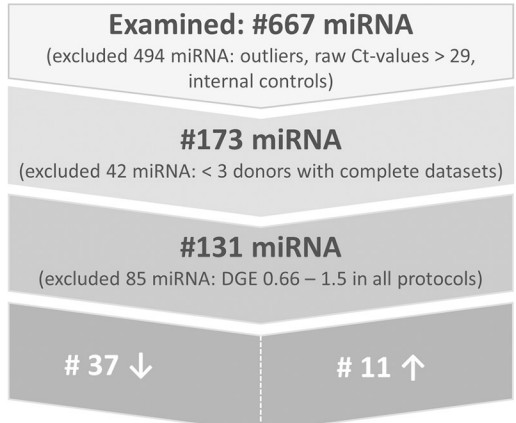

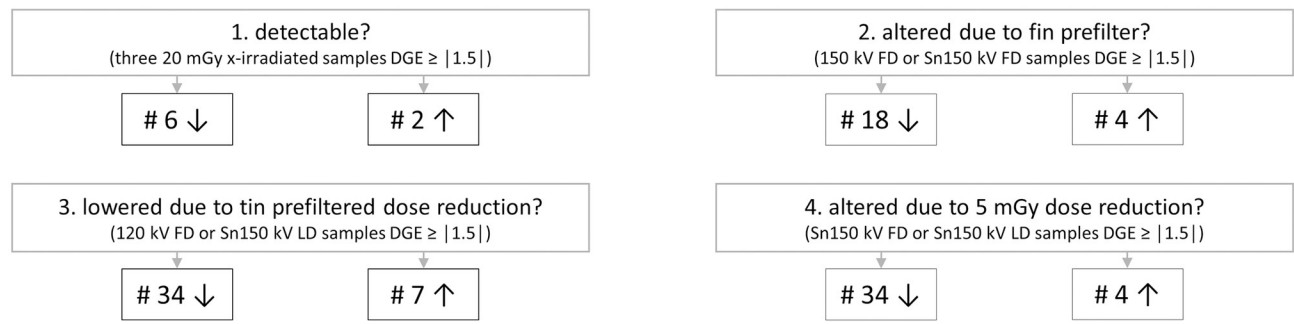

**Fig 5. Overview of miRNA evaluation sequence.** In the first step (upper half) candidate miRNAs were identified. Candidate miRNAs showed a DGE ≥ |1.5| in the samples from at least 3 donors in at least one protocol. In a second step (lower half), the four research questions (radiation-induced DGE, 150 kV FD vs. Sn 150 kV FD, 120 kV FD vs. Sn 150 kV LD, Sn 150 kV FD vs. Sn 150 kV LD) were further evaluated, resulting in 8 to 41 potentially dysregulated miRNAs per research question. The selection criteria to detect dysregulated miRNA was intentionally set low (min. n = 3 donors, DGE |1.5|) to avoid premature rejection of candidate miRNA in this pilot study. Abbreviations: Ct = cycle threshold, DGE = differential gene expression, FD = full dose, kV = kilovolt, LD = low dose, miRNA = microRNA, Sn = tin prefilter.

samples. No significant change ($p$ = 0.21–1.00) in DGE was seen when comparing the Sn 150 kV FD with the 5 mGy lesser X-irradiated Sn 150 kV LD protocol in the 38 miRNAs, which showed up- or down-regulation in either the Sn 150 kV FD or the Sn 150 kV LD protocols.

## Discussion

There is a constant increase in the number of annual computed tomography (CT) examinations which causes the majority of the dose contribution of medical radiation exposure to the population [1, 2]. Hence, it is crucial to develop and use dose-reducing methods, such as the tin prefilter (Sn) technique, to keep radiation dose as low as reasonably achievable (ALARA). Furthermore, there is increasing interest to investigate the molecular biological effects of low-level radiation. Gene expression (GE) analysis of known radiation-induced genes and determination of induced DNA DSBs by means of γ-H2AX and 53BP1 foci enumeration are the two radiobiological assays that can detect radiation exposures in the mGy range and are, therefore, most commonly used for this purpose [20, 25, 30–32]. Furthermore, miRNA levels in tissues and blood samples are known to be influenced by ionizing radiation exposures in animal models [33–35] and in humans [36, 37] and their expression levels can distinguish between whole-body and partial-body irradiation [33]. miRNA are single-stranded, non-coding RNA

molecules that are involved in the regulation of protein production by either inhibiting mRNA translation or leading to mRNA degradation [38–40]. So far, miRNA expression after low-level irradiation has not yet been extensively investigated.

In this study, we used EDTA whole blood samples from 6 healthy donors. We irradiated them *ex-vivo* in a CT scanner with or without tin prefiltration using full dose (FD) protocols (120 kV, 150 kV, Sn 150 kV, 20–22 mGy) and a tin prefiltered low dose (LD) protocol (Sn 150 kV, 16.5 mGy) to investigate the influence of the tin prefilter on radiation-induced DNA DSBs and on GE changes of known radiation-induced genes such as *FDXR*, *DDB2*, *EDA2R*, *AEN*, *CDKN1A*, *BAX* and *APOBEC3H*, as well as on GE changes of 667 of 1917 identified miRNA. The CT protocols used allowed the following four questions to be examined: Are radiation-induced molecular biological changes 1) detectable (sham vs. irradiated), 2) altered due to tin prefiltration (150 kV FD vs Sn 150 kV FD), 3) lowered due to tin prefiltered dose reduction (120 kV FD vs Sn 150 kV LD, same image quality) and 4) is a 5 mGy dose reduction detectable with current biodosimetry assays (Sn 150 kV FD vs Sn 150 kV LD)?

Relative to unexposed samples, we identified radiation-induced DNA DSBs and GE changes of *DDB2* and *EDA2R*, but no miRNA changes and no GE changes of *AEN*, *CDKN1A*, *BAX*, and *APOBEC3H* could be detected after low-level irradiation of 16 to 20 mGy. Significant radiation-induced *FDXR* differential gene expression (DGE) was observed with the FD protocols but not with the tin-prefiltered LD protocol. There were no significant changes in DNA DSBs and DGE of mRNA and miRNA when the tin prefilter was used (150 kV FD vs. Sn 150 kV FD protocol), when the dose was reduced (~ 3.5 mGy) due to tin prefiltering while maintaining image quality (120 kV vs Sn 150 kV LD protocol), or when trying to detect a dose reduction of 5 mGy (Sn 150 kV FD vs Sn 150 kV LD). However, in a frequency distribution comparing the Sn FD protocol with the Sn LD protocol and the 120 kV FD protocol with the Sn150 kV LD protocol, the Sn LD protocol showed a significantly lower frequency of lower DGE compared to the FD protocols. Together with the *FDXR* results, the DGE frequency distribution results indicate a tendency toward lower molecular biological cell response in tin-prefiltered LD protocols compared to the 120 kV FD protocol with the same image quality. Due to the non-significant changes in the direct group comparisons, these results of the frequency distribution should not be overinterpreted, so that the overall conclusion of this study, that the tin prefilter does not lead to any relevant change in the analyzed biological endpoints, remains valid. The results of the frequency distribution are nevertheless worth mentioning, as they show the protective, expected effect of the tin prefilter technique due to the dose reduction while maintaining image quality.

GE changes of *FDXR* and *DDB2* have been intensively investigated after high radiation exposure [41–44], but it has also been described in the few existing studies that they are dysregulated after low radiation exposure [20, 25, 32, 45]. Only recently, it was shown that EDA2R was strongly dysregulated even after low dose radiation exposure (2.5 mGy) and is, therefore, a promising biomarker to detect radiation-induced changes in the mGy range [20, 25]. It could, therefore, also be used to investigate possible effects of dose reduction techniques on the transcriptome. Nonetheless, *EDA2R*, like all other genes, was unable to detect the 3.5 to 5.5 mGy dose reduction of the Sn 150 kV LD protocol, although *EDA2R* showed the highest radiation-induced DGE levels. Possible reasons include the high interindividual variance previously described for *EDA2R* [46] and that current biodosimetry assays cannot detect a dose reduction of 3.5 to 5.5 mGy and the difference induced by tin prefiltering.

Although trial CT scans were performed to determine the acquisition parameters of the scanner to ensure 20 mGy for FD scans and the same image quality as the 120 kV FD scan for the Sn LD scan, the retrospective evaluation of the thermoluminescence dosimeter (TLD) data and the image quality shows deviations in the targeted dose as well as the image quality

(Table 1). This is a limitation of this study, which was attempted to be minimized by performing the trial scans. Another explanation for why no differences in molecular biological changes were observed when using tin prefiltration is the geometry of the radiation exposure itself by positioning the blood samples centrally in the water phantom and not at the periphery, where larger differences would have been expected. Other limitations include the low number of donors examined which may lead to higher statistical effects due to more data scattering because of interindividual variations. Furthermore, the results were obtained by *ex-vivo* analysis, which will require further investigation with *in-vivo* samples and larger case numbers are required to verify our results.

Despite the studies' limitations, our data indicate that 1) there are no radiation-induced miRNA changes in the dose range applied by CT scans, 2) the tin prefilter technique does not prompt significant radiation-induced GE changes and changes in the number of DNA DSB. Still, our GE results suggest a tendency toward less radiation-induced GE changes of the tin prefiltered protocol with a 3.5 mGy dose reduction, while the image quality remained the same or was even better than the 120 kV protocol. Therefore, the tin prefilter technique can safely be applied without eliciting further DNA damage or changes at the transcriptional and the majority of the post-transcriptional level. Our data suggest, that the tendency of tin-filtering protocols to induce lower molecular biological changes is linked to the reduction in absorbed dose and a shift toward higher radiation energy levels.

## Supporting information

**S1 Fig.** Picture of the 32-cm water phantom within the CT scanner (A) and CT-image (B) of the same water phantom, filled with 37˚C tempered water and ETDA-blood tubes positioned in the center. The elongated light gray line measuring a few mm directly above the blood samples is the TLD positioned next to the blood samples. Two additional tubes filled with different contrast media concentrations are positioned on the side, which were not used for this study. Abbreviations: EDTA = ethylenediaminetetraacetic acid, TLD = thermoluminescent dosimeter.
(TIF)

**S2 Fig. Immunofluorescence microscope images of (co)-localizing γ-H2AX + 53BP1 foci for unexposed and exposed samples of one donor.** Example images show 20–21 peripheral blood cells (blue) per image with visualisation of γH2AX (green), 53BP1 (red) and co-localizing γ-H2AX + 53BP1 (yellow) double-strand breaks (DSB). There is an increase in DNA DSBs in X-irradiated samples. Abbreviations: FD = full dose, kV = kilovolt, LD = low dose, Sn = tin prefilter.
(TIF)

**S3 Fig. Frequency distribution results of excess radiation-induced foci.** Graphical (connected scatter plot) and tabular (chi-square table) representation of the frequency distribution of higher resp. lower excess radiation-induced γH2AX Foci for the following research questions: A) 150 kV FD vs. Sn 150 kV FD, B) 120 kV FD vs. Sn 150 kV LD and C) Sn 150 kV FD vs. Sn 150 kV LD. The symbols reflect the mean values of excess radiation-induced foci per category. The line connects the excess radiation-induced Foci results of the same donor for both compared groups. Abbreviations: FD = full dose, kV = kilovolt, LD = low dose, Sn = tin prefilter.
(TIF)

**S4 Fig. Scatter plot of differential gene expression of additional *FDXR and DDB2 Taq-Man assays* (relative to unexposed) for each examined CT scan protocol.** These *Taq-Man*

*assays* were previously identified as most suitable for biodosimetry purposes [28]. The symbols reflect mean values, and error bars represent the standard error of the mean. P-values <0.05– 0.01 and <0.01–0.001 are marked with one or two asterisks and refer to significant differences relative to unexposed values. Abbreviations: FD = full dose, kV = kilovolt, LD = low dose, Sn = tin prefilter.
(TIF)

**S1 Table. Raw data of transcriptional qRT-PCR results and γH2AX DNA double-strand break analyses.** GE raw data (raw Ct-values, 18S rRNA normalized Ct-values, differential gene expression relative to unexposed), as well as γH2AX DNA double-strand break analyses of raw data (average foci per cell, radiation-induced foci (RIF)) per donor and protocol, are presented. Abbreviations: Abs.Div. = absolute difference, av. = average, Ct = cycle threshold, DGE = differential gene expression, Std. Dev. = standard deviation, SE = standard error.
(XLSX)

**S2 Table. 8–41 candidate miRNAs listed per research question with name, up- or downregulation, number of donors in which miRNA was detected, DGE mean value and standard deviation, and exact statistical test results.** Abbreviations: DGE = differential gene expression, FD = full dose, LD = low dose, miRNA = micro RNA, Sn = tin prefilter, std. dev. = standard deviation, # = number.
(XLSX)

## Author Contributions

**Conceptualization:** Simone Schüle, Carsten Hackenbroch, Patrick Ostheim, Cornelius Hermann, Harry Scherthan, Michael Abend.

**Data curation:** Simone Schüle, Razan Muhtadi, Samantha Stewart, Michael Abend.

**Formal analysis:** Simone Schüle, Patrick Ostheim, Cornelius Hermann, Michael Abend.

**Investigation:** Simone Schüle, Carsten Hackenbroch, Razan Muhtadi, Harry Scherthan.

**Methodology:** Simone Schüle, Patrick Ostheim, Harry Scherthan, Michael Abend.

**Project administration:** Simone Schüle, Meinrad Beer, Matthias Port, Michael Abend.

**Resources:** Carsten Hackenbroch, Meinrad Beer, Matthias Port, Michael Abend.

**Supervision:** Meinrad Beer, Matthias Port, Michael Abend.

**Visualization:** Simone Schüle, Razan Muhtadi, Samantha Stewart, Harry Scherthan, Michael Abend.

**Writing – original draft:** Simone Schüle, Michael Abend.

**Writing – review & editing:** Simone Schüle, Carsten Hackenbroch, Meinrad Beer, Patrick Ostheim, Cornelius Hermann, Razan Muhtadi, Samantha Stewart, Matthias Port, Harry Scherthan.

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
