## [Decision Letter · Decision Letter 0]

14 Oct 2024

PONE-D-24-34482Tin prefiltration in computed tomography fails to alter radiation-induced gene expression and double-strand break formationPLOS ONE

Dear Dr. Abend,

Thank you for submitting your manuscript to PLOS ONE. After careful consideration, we feel that it has merit but does not fully meet PLOS ONE’s publication criteria as it currently stands. Therefore, we invite you to submit a revised version of the manuscript that addresses the points raised during the review process.

We look forward to receiving your revised manuscript.

Kind regards,

Minsoo Chun, Ph.D.

Academic Editor

PLOS ONE

Reviewers' comments:

Reviewer's Responses to Questions

**Comments to the Author**

1. Is the manuscript technically sound, and do the data support the conclusions?

Reviewer #1: Yes

Reviewer #2: Yes

2. Has the statistical analysis been performed appropriately and rigorously? 

Reviewer #1: Yes

Reviewer #2: Yes

3. Have the authors made all data underlying the findings in their manuscript fully available?

Reviewer #1: Yes

Reviewer #2: Yes

4. Is the manuscript presented in an intelligible fashion and written in standard English?

Reviewer #1: Yes

Reviewer #2: Yes

5. Review Comments to the Author

Reviewer #1: General Comments

The manuscript titled "Tin prefiltration in computed tomography fails to alter radiation-induced gene expression and double-strand break formation" presents an interesting study on the molecular biological response to the tin prefilter technique in CT. The authors examine both radiation-induced double-strand break (DSB) formation and differential gene expression (DGE) in blood samples exposed to different CT protocols. The study is relevant in the context of reducing radiation dose without compromising image quality, and the results indicate that this technique does not significantly affect radiation-induced molecular changes when compared to standard CT protocols.

Specific Comments:

1. Abbreviations, Units, and Sentence Structure:

The entire manuscript requires a thorough revision to ensure consistent use of abbreviations, correct application of units in the biological endpoints measured, and the construction of complete sentences.

2. Image Quality Technique:

The description of image quality techniques should be more detailed and clearer. It is essential to specify whether the quality of the images was evaluated objectively or subjectively.

3. Title Suggestion:

I recommend revising the title to one of the following options for clarity and impact:

- Tin prefiltration in computed tomography does not significantly alter radiation-induced gene expression and DNA double-strand breaks."

- Minimal effects of tin prefiltration on radiation-induced gene expression and DNA double-strand break formation in computed tomography."

4. Abstract Revision (Page 3):

In the conclusion of the abstract, instead of "our ex vitro," it is better to write "the current ex-vivo study."

5. miRNA Notation:

The first miRNA should be correctly noted as "MicroRNA-667."

6. Dose Measurement Methodology:

The methodology section should clearly explain how the dose was measured or calculated, whether through dosimetry or other methods.

7. Control Blood Samples:

The exclusion criteria for the healthy control blood samples should be specified. It would have been better to compare the same donor blood samples before and after irradiation. If controls excluded certain criteria such as smoking or chemotherapy, this should be clearly stated.

8. Discussion on DGE Values:

Although the paper concludes that tin prefiltration does not alter radiation-induced DGE or DSB formation, it also notes a significant tendency toward lower DGE values for the Sn 150 kV low-dose protocol. This finding needs clarification. The discussion should be expanded to better explain the biological implications of this trend toward lower DGE values. Additionally, it should be explicitly stated whether these changes, though not statistically significant, could still hold clinical relevance.

Reviewer #2: The authors have presented a careful study of an assessment of the tin pre-filtration technique for CT scanning. This technique has the ability to reduce radiation exposure during CT scans by eliminating the lower energy radiation from the scanners which gets absorbed by a thin layer of tin (Sn), allowing only the higher energy radiation. This is best suited for deep and high contrast imaging. A thorough evaluation of the possible protective effect of this technology is significant and this reviewer believes it has been done carefully, and with suitable sampling (at least for the DNA DSB marker yH2AX), and with appropriate assays. The authors chose to concentrate on the DNA damage aspects of CT scanning. This reviewer believes this study was straightforward, and presents reliable data that is easily decipherable and would be of value to the field. Care was also taken to control for dose levels, and several iterations were used for a full comparison. It was also commendable that only yH2AX foci co-localized with 53BP1 were counted, and yH2AX is known to mark non-DSB lesions as well. The number of patient/samples (6) could have been higher, which might have helped reducing variability in the transcriptomics and miRNA (but that was acknowledge by the authors).

some minor edits:

line 242 Fig 2. yH2AX foci: the scoring and tabulating methodologies for this assay are sound. according to the method to score these the number of foci per cell of unexposed is subtracted from the exposed. The resulting increase in foci count is not that high, it is less than 1 foci per cell, but nevertheless significant. This reviewer would be heartened to see some example images. The increase in foci might not be perceivable, but showing example imaging (even if in the supplement) would boost confidence in the technique.

line 370-371

suggested change to

"can distinguish between whole-body and partial-body irradiation [33]."

line 374

suggested change to

"low dose irradiation has not been"

line 403 and 404

change deregulated to 'dysregulated'

line 407

change genome to 'transcriptome'

line 415

suggested change to

"This is a limitation of this study, and which was attempted to be minimized by performing the trial scans."

6. PLOS authors have the option to publish the peer review history of their article (what does this mean?). If published, this will include your full peer review and any attached files.

Reviewer #1: **Yes: **Hariwan A. Mohammed

Reviewer #2: **Yes: **Aris A. Polyzos

---

## [Author Response · Author response to Decision Letter 0]

14 Nov 2024

Dear Editor and Reviewers,

thank you for your valuable comments. Please find our point-by-point response in the attached file "Response to Reviewers".

Kind regards

---

## [Decision Letter · Decision Letter 1]

3 Dec 2024

Tin prefiltration in computed tomography does not significantly alter radiation-induced gene expression and DNA double-strand break formation

PONE-D-24-34482R1

Dear Dr. Abend,

We’re pleased to inform you that your manuscript has been judged scientifically suitable for publication and will be formally accepted for publication once it meets all outstanding technical requirements.

Kind regards,

Minsoo Chun, Ph.D.

Academic Editor

PLOS ONE

Additional Editor Comments (optional):

Reviewers' comments:

Reviewer's Responses to Questions

**Comments to the Author**

1. If the authors have adequately addressed your comments raised in a previous round of review and you feel that this manuscript is now acceptable for publication, you may indicate that here to bypass the “Comments to the Author” section, enter your conflict of interest statement in the “Confidential to Editor” section, and submit your "Accept" recommendation.

Reviewer #1: All comments have been addressed

Reviewer #2: All comments have been addressed

2. Is the manuscript technically sound, and do the data support the conclusions?

Reviewer #1: Yes

Reviewer #2: Yes

3. Has the statistical analysis been performed appropriately and rigorously? 

Reviewer #1: Yes

Reviewer #2: Yes

4. Have the authors made all data underlying the findings in their manuscript fully available?

Reviewer #1: Yes

Reviewer #2: Yes

5. Is the manuscript presented in an intelligible fashion and written in standard English?

Reviewer #1: Yes

Reviewer #2: Yes

6. Review Comments to the Author

Reviewer #1: (No Response)

Reviewer #2: The authors responded very thoroughly to the initial review comments provided for the manuscript.

Including figures of the quantified data for DNA damage (imaging of gamma-H2AX nuclear foci) allows a reader or investigator trying to recapitulate these findings to properly evaluate both the data and the quality of the signal. This was the chief issue this reviewer had with the original manuscript. Otherwise, it is very thoroughly detailed and annotated and in this reviewers opinion a worthy publication for the community.

7. PLOS authors have the option to publish the peer review history of their article (what does this mean?). If published, this will include your full peer review and any attached files.

Reviewer #1: **Yes: **Hariwan Abdulkareem Mohammed

Reviewer #2: **Yes: **Aris Polyzos

---

## [Editor Report · Acceptance letter]

6 Dec 2024

PONE-D-24-34482R1 

PLOS ONE

Dear Dr. Abend, 

I'm pleased to inform you that your manuscript has been deemed suitable for publication in PLOS ONE. Congratulations! Your manuscript is now being handed over to our production team.

Kind regards, 

on behalf of

Dr. Minsoo Chun 

Academic Editor

PLOS ONE